# Temperature-Dependent Evolution of Raman Spectra of Methylammonium Lead Halide Perovskites, CH_3_NH_3_PbX_3_ (X = I, Br)

**DOI:** 10.3390/molecules24030626

**Published:** 2019-02-11

**Authors:** Kousuke Nakada, Yuki Matsumoto, Yukihiro Shimoi, Koji Yamada, Yukio Furukawa

**Affiliations:** 1Department of Chemistry and Biochemistry, School of Advanced Science and Engineering, Waseda University, Shinjuku, Tokyo 169-8555, Japan; sw-ko@asagi.waseda.jp; 2Research Center for Computational Design of Advanced Functional Materials (CD-FMat), National Institute of Advanced Industrial Science and Technology (AIST), 1-1-1 Umezono, Tsukuba, Ibaraki 305-8568, Japan; matsumoto.yuki2@mazda.co.jp (Y.M.); y.shimoi@aist.go.jp (Y.S.); 3College of Industrial Technology, Nihon University, Izumi-cho 1-2-1, Narashino, Chiba 27-8575, Japan; yamada.kouji@nihon-u.ac.jp

**Keywords:** Raman spectroscopy, phase transition, motional narrowing, bandwidth, organic/inorganic hybrid perovskites, methylammonium lead halide perovskites

## Abstract

We present a Raman study on the phase transitions of organic/inorganic hybrid perovskite materials, CH_3_NH_3_PbX_3_ (X = I, Br), which are used as solar cells with high power conversion efficiency. The temperature dependence of the Raman bands of CH_3_NH_3_PbX_3_ (X = I, Br) was measured in the temperature ranges of 290 to 100 K for CH_3_NH_3_PbBr_3_ and 340 to 110 K for CH_3_NH_3_PbI_3_. Broad ν_1_ bands at ~326 cm^−1^ for MAPbBr_3_ and at ~240 cm^−1^ for MAPbI_3_ were assigned to the MA–PbX_3_ cage vibrations. These bands exhibited anomalous temperature dependence, which was attributable to motional narrowing originating from fast changes between the orientational states of CH_3_NH_3_^+^ in the cage. Phase transitions were characterized by changes in the bandwidths and peak positions of the MA–cage vibration and some bands associated with the NH_3_^+^ group.

## 1. Introduction

Organic/inorganic hybrid perovskite materials, CH_3_NH_3_PbX_3_ (X = I, Br), have attracted much research interest for their use in solar cells over the past few years because solar cells fabricated with perovskite show high power conversion efficiencies of more than 20% [1,2,3]. Although perovskites form a large family of compounds with ABX_3_ stoichiometry, the peculiar structure of the materials is a combination of organic and inorganic groups with a unique interplay, which influences their optical and electronic properties. CH_3_NH_3_PbI_3_ exhibits a broad absorption from visible to near-infrared [1,4]; its band gap is 1.52 eV [5]. The electron-hole diffusion length is ~100 nm [6,7]. Charge carriers exhibit high mobilities [5,8]. The excellent power conversion efficiencies of the organic/inorganic hybrid perovskite solar dells originate from these properties.

The crystal structures and phase transitions of CH_3_NH_3_PbX_3_ (X = I, Br) were determined by X-ray and neutron diffraction [9,10,11,12], calorimetric measurements [12,13], and nuclear quadrupole resonance spectroscopy [14]. The crystal structures and transition temperatures are listed in Table 1. There are four and three crystal phases in CH_3_NH_3_PbBr_3_ and CH_3_NH_3_PbI_3_, respectively. In one study [9], the space group of the orthorhombic phases was found to be *Pna*2_1_. Later reinvestigations of the space group [10,11] indicated that the space group is *Pnma*. In these crystal structures, there exists a methylammonium (MA) CH_3_NH_3_^+^ cation at the center of a cube formed by corner-sharing PbX_6_ octahedra, i.e., an anionic PbX_3_ network. All the phase transitions are of the order-disorder type [13] and of the first order, although the highest-temperature transitions are close to the second order [13]. In diffraction studies, the position, orientation, and rotation around the C−N bond of the MA cation within the inorganic network were not determined completely [9,10,11,12,13,14]. MA cations have dynamically disordered states in orientation of the C−N axis and around the C−N axis [13,15,16,17,18]. The reorientational dynamics of MA cations were investigated [15,16,17,18]. The characteristic reorientation time was less than a few ps [16,17]. The electrostatic interaction between positive MA cations and the anionic PbX_3_ network, which is governed by hydrogen-bonding interactions between the NH_3_^+^ group in the MA cation and the electronegative halide atoms [17,18], will play an important role in the dynamics of MA cations. The electronic structures of organic/inorganic hybrid perovskites were calculated [19,20,21,22,23,24,25]. The optical band gap originates from a direct transition between Pb(6s)–I(5p) valence bands and Pb(6p) conduction bands. The optical band gap strongly correlates with the bond angles through the steric size of the molecular cation existing in the inorganic PbX_3_ network [23]. The interaction of the organic cation with the PbX_3_ network also has an impact on photoluminescence [26,27,28], exciton binding energy [29], and charge carrier lifetime [30].

Infrared and Raman spectroscopies are very useful for studying phase transitions, lattice dynamics, and interactions between the organic and inorganic counterparts of hybrid perovskite materials. In particular, Raman spectroscopy can be used for in situ studies of devices fabricated on a glass substrate, i.e., buried layers. Onoda-Yamamuro [13] reported that an infrared band assigned to MA rocking at ~910 cm^−1^ is sensitive to the phase transitions of MAPbX_3_ (X = I, Br, Cl) and determined the correlation times and activation energies for the hindered rotational motion of MA from the widths of the band. After the report of high-performance solar cells fabricated with MAPbI_3_ [1], many infrared [31,32,33,34,35] and Raman [36,37,38,39,40,41,42,43] studies were performed. Low-wavenumber lattice vibrations can be detected by Raman spectroscopy. However, it should be noted that MAPbI_3_ compounds are damaged under strong laser irradiation, even in vacuum, although these compounds are unstable in ambient air [37]. The observed infrared and Raman bands were assigned on the basis of the results of theoretical calculations [33,36,38,40,41,43,44]. The observed bands can be attributed to lattice vibrations and intramolecular vibrations of MA. The lattice vibrations were analyzed [41,43]. The reorientation dynamics of MA ions in MAPbI_3_ was studied by two-dimensional (2D) infrared spectroscopy and molecular dynamics [32]. The characteristic reorientation time was ~300 fs and ~3 ps [32]. The temperature evolution of infrared spectra [35] and Raman spectra [39,40,41] was investigated in order to characterize phase transitions and vibrational relaxation. However, detailed temperature dependence of Raman bands was necessary in order to further understand the phase transitions and the interaction between the organic cation and the PbX_3_ network.

In this study, we investigated the temperature-dependent evolution of almost of all the Raman bands for MAPbI_3_ and MAPbBr_3_ in a wide range of temperature. We analyzed the changes in Raman bandwidths and peak positions across the phase transition temperatures in detail, focusing our attention on the intramolecular vibrations of MA. We proposed motional narrowing for broad bands at ~326 cm^−1^ for MAPbBr_3_ and at ~240 cm^−1^ for MAPbI_3_. We here discuss the rotational dynamics of MA cations, which is closely related to the optical properties of the perovskites.

## 2. Results and Discussion

### 2.1. Temperature-Dependent Evolution of the Raman Spectrum of MAPbBr_3_

The Raman spectra of an MAPbBr_3_ pellet at 100, 150, 200, and 290 K with an excitation wavelength of 633 nm are shown in Figure 1. The Raman spectra at 100, 150, 200, and 290 K originate from the orthorhombic, tetragonal I, tetragonal II, and cubic phases, respectively.

Observed Raman bands below 200 cm^−1^ are attributed to lattice vibrations [40,41]. The lattice vibrations of the orthorhombic phase were assigned [41]. The bands above 400 cm^−1^ are attributed to intramolecular vibrations of MA. The assignments of the observed MA bands made on the basis of those reported in previous papers [41,43,44,45] are listed in Appendix A.

A band observed at 328–325 cm^−1^ is called ν_1_. Spectral features of the ν_1_ band are peculiar. This band is very broad compared with other bands. This band is attributed to the torsional vibration of MA [40,41], and the motion is coupled with the inorganic cage through NH···Br hydrogen bonds [41]. Similar modes were observed at 240–249 cm^−1^ for MAPbI_3_ [38,41] and at 488 cm^−1^ for MAPbCl_3_ [41,46]; the observed peak position depends largely on X, which is consistent with the coupling through the NH···X hydrogen bonds. The intensity of the band is stronger than those of the intramolecular vibrations at 150, 200, and 290 K. For a free MA molecule, the torsional band should be much less intense than those attributed to intramolecular vibrations. Recently, Mattoni et al. [44] reported the theoretical temperature evolution of vibrational spectra of MAPbI_3_ using density functional theory calculations and classical molecular dynamics. They found that the thermally induced weakening of the H···I interactions and the anharmonic mixing of modes give two vibrational peaks at 200–250 cm^−1^ (X band) that are not present below 10 K. The X band is not a pure torsion (twisting), but mixed with breathing or rocking motions of hydrogen atoms. The observed ν_1_ band is attributable to the theoretical X band. We call this band the MA–cage vibration. The assignments of other main bands are as follows: ν_2_, ~917 cm^−1^, CH_3_ rocking and NH_3_^+^ rocking; ν_3_, ~971 cm^−1^, C−N^+^ stretching; ν_8_, ~1478 cm^−1^, NH_3_^+^ symmetric deformation; ν_9_, ~1592 cm^−1^, NH_3_^+^ degenerate deformation; ν_12_, ~2965 cm^−1^, CH_3_ symmetric stretching. The ν_2_, ν_8_, and ν_9_ bands are associated with the NH_3_^+^ group.

The widths (full width at half maximum, FWHM) and the peak positions of the ν_1_, ν_2_, ν_3_, ν_8_, ν_9_, and ν_12_ bands are shown in Figure 2. Across the orthorhombic-to-tetragonal I phase transition at 149 K, the FWHMs of the bands show large changes—the ν_2_ and ν_9_ bands exhibit abrupt increases, whereas the ν_1_, ν_3_, ν_8_, and ν_12_ bands exhibit sharp bends. The peak positions of the ν_2_, ν_3_, and ν_8_ bands show apparent abrupt increases. These bandwidth and peak position results are consistent with a first-order phase transition. The ordering of MA cations is associated with this transition.

Across the transition from the tetragonal I phase to tetragonal II phase at 154 K, the FWHM of the ν_9_ band exhibits an apparent abrupt increase, although the other bands exhibit no apparent changes. The peak position of the ν_1_ band exhibits a small deflection. This abrupt change is consistent with a first-order phase transition. It is interesting that only the ν_9_ band assigned to NH_3_^+^ degenerate deformation exhibits a large FWHM change.

Across the transition from the tetragonal II phase to the cubic phase at 236 K, the FWHM of the ν_2_ band shows a significant deflection. The peak positions of the ν_1_ and ν_8_ bands exhibit small but abrupt decreases, whereas the peak position of the ν_3_ band exhibits a small but abrupt increase. The peak position of the ν_2_ band exhibits a dispersion-like feature. The observed changes across the tetragonal II-to-cubic transition are small. These small changes indicate that this transition is close to the second order.

The FWHM of the ν_1_ band decreases noticeably at the orthorhombic, tetragonal I, and tetragonal II phases except at 130–150 K, whereas the widths of other bands exhibit increases or plateaus. This is an anomalous temperature dependence, which is attributable to the motional narrowing phenomenon in the nuclear magnetic resonance theory [47]. The MA ion in the PbX_3_ cage can take several orientations [13,16,17,18]. Each orientation gives rise to a different vibrational state, i.e., peak position in the Raman spectrum. If the reorientation change of MA cations is fast, motional narrowing can be observed. The change in the width of the ν_1_ band is attributable to rapid exchanges among these orientational states.

### 2.2. Temperature Evolution of the Raman Spectrum of MAPbI_3_

The Raman spectra of an MAPbI_3_ pellet at 120, 170, and 340 K with 830 nm excitation are shown in Figure 3. In these spectra, CH stretching bands were not observed because the charge-coupled device detector has no spectral response in this region. MAPbI_3_ exhibits photoluminescence at 776 nm [22]. The tail of the photoluminescence was observed as a background with Raman bands. The intensity of photoluminescence increased with increasing temperature. The Raman spectra at 120, 170, and 340 K originate from the orthorhombic, tetragonal, and cubic phases, respectively. Raman bands below 200 cm^−1^ are attributed to lattice vibrations, whereas the bands above 400 cm^−1^ are attributed to intramolecular vibrations of MA [41,43,44]. The assignments of the intramolecular vibrations are listed in Appendix A.

Spectral features of the ν_1_ band observed at 240–249 cm^−1^ are peculiar, as described for MAPbBr_3_ in 2.1. This band is very broad compared with other bands. The ν_1_ band is assigned to the MA–cage vibration, corresponding to the X band in the theoretical study [44]. The band originates from the thermally induced weakening of the H···I interactions [44]. As shown in Figure 3, the relative intensity of the ν_1_ band increases with increasing temperature, which is consistent with the temperature evolution of the X band [44]. At temperatures lower than 140 K, the shape of the ν_1_ band showed changes. Thus, the ν_1_ band was decomposed with three bands at approximately 265, 243, and 225 cm^−1^ by least-squares curve fitting. In a previous paper [41], five peaks at 312, 272, 243, 223, and 199 cm^−1^ were reported for MAPbI_3_. In the orthorhombic phase, MA cations are fully ordered [18]. Thus, the observed spectral feature are attributed to this ordering of MA cations.

The assignments of other main bands are as follows: ν_2_, ~912 cm^−1^, CH_3_/NH_3_^+^ rocking; ν_3_, ~964 cm^−1^, C−N^+^ stretching; ν_8_, ~1466 cm^−1^, NH_3_^+^ symmetric deformation; ν_9_, ~1584 cm^−1^, NH_3_^+^ degenerate deformation.

The FWHMs and peak positions of the ν_1_, ν_2_, ν_3_, ν_8_, and ν_9_ bands are shown in Figure 4. Across the orthorhombic-to-tetragonal phase transition at 161 K, the FWHMs of the bands show large changes: the ν_2_, ν_3_, ν_8_, and ν_9_ bands exhibit abrupt increases, while the ν_1_ band exhibits an abrupt decrease. The ν_2_ and ν_9_ bands shift downward, and the ν_1_ and ν_8_ bands shift upward. In the temperature range between 150 and 160 K, weak bands associated with the tetragonal phase are observed, which is probably attributed to a supercooled state with the decreasing temperature. These changes in bandwidth and peak position are consistent with a first-order phase transition.

Across the tetragonal-to-cubic phase transition at 330 K, the FWHM of the ν_1_ band exhibits an abrupt but small decrease, although the FWHMs of other bands show no abrupt changes or deflections. None of the bands show significant wavenumber shifts. No significant changes of the bandwidths and peak positions of the infrared and Raman bands were found across the tetragonal-to-cubic phase transition of MAPbI_3_ [35,40]. The observed small change indicates that this transition is close to the second order.

The FWHM of the ν_1_ band decreases over the whole temperature range, whereas the FWHMs of other bands increase or plateau. This anomaly is attributable to motional narrowing [47], as described in Section 2.1. The MA ion in the PbX_3_ cage can take several orientations [13,16,17,18]. Motional narrowing was also observed for the ν_1_ band of MAPbBr_3_, as described above. However, it is interesting that the FWHM of a torsional vibration at ~480 cm^−1^ for MAPbCl_3_ increases with increasing temperature in the range of 77 to 280 K [46].

## 3. Materials and Methods

Samples of MAPbBr_3_ and MAPbI_3_ crystal powders were prepared according to previous papers [14,48]. Compressed micro pellets of the crystalline powders were made. A pellet was placed on a cold head of a liquid-nitrogen-cooled cryostat (Oxford Instruments, Oxon, UK, DN1754) equipped with a temperature controller (Oxford Instruments, ITC5025). The Raman spectra of the pellets were measured in the backscattering configuration on a Raman spectrometer (Renishaw InVia, Gloucestershire, UK) using a macro sampling set with a 60-mm focal length lens (numerical aperture, 0.08). The Raman spectra of the MAPbBr_3_ pellet were measured with excitation at 633 nm and a power of ~500 μW in the temperature range from 290 to 100 K. The Raman spectra of the MAPbI_3_ pellet were measured with excitation at 830 nm and a power of ~900 μW in the temperature range from 340 to 110 K. We obtained the peak wavenumber of the FWHM of a band using the Renishaw Wire 3.1 program. An observed band was fitted with a linear combination of the Gaussian and Lorentzian functions and the removal of a baseline by the least-squares curve fitting. The errors of the peak wavenumber and FWHM were ±0.5 and ±1 cm^−1^, respectively. No error bars were provided for simplicity.

## 4. Conclusions

The temperature-dependent evolution of the Raman spectra of the organic/inorganic hybrid perovskite MAPbX_3_ (X = I, Br) was measured. Broad ν_1_ bands at ~326 cm^−1^ for MAPbBr_3_ and at ~240 cm^−1^ for MAPbI_3_ were assigned to the MA–PbX_3_ cage vibrations activated by the thermally induced weakening of the H···X interactions. These bands exhibited anomalous temperature dependence, which was attributable to motional narrowing originating from fast changes between the orientational states of CH_3_NH_3_^+^ in the PbX_3_ cage. Phase transitions were characterized by changes in the bandwidths and peak positions of the MA–cage vibrations and some bands were associated with the NH_3_^+^ group. Across the orthorhombic-to-tetragonal phase transition, abrupt and large increases or decreases in the bandwidth and peak position were observed, which is consistent with a first-order transition. However, across the tetragonal-to-cubic phase transition, small changes were observed.

## Figures and Tables

**Figure 1 molecules-24-00626-f001:**
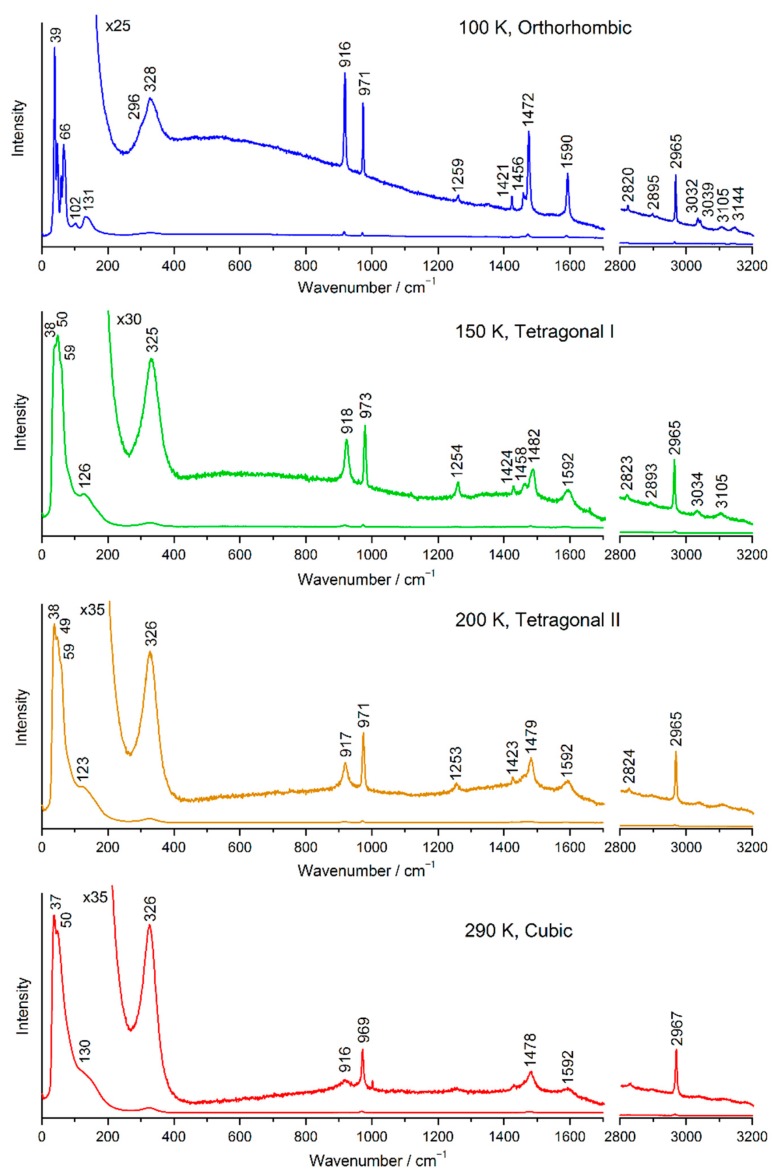
Raman spectra of an MAPbBr_3_ pellet; excitation wavelength: 633 nm.

**Figure 2 molecules-24-00626-f002:**
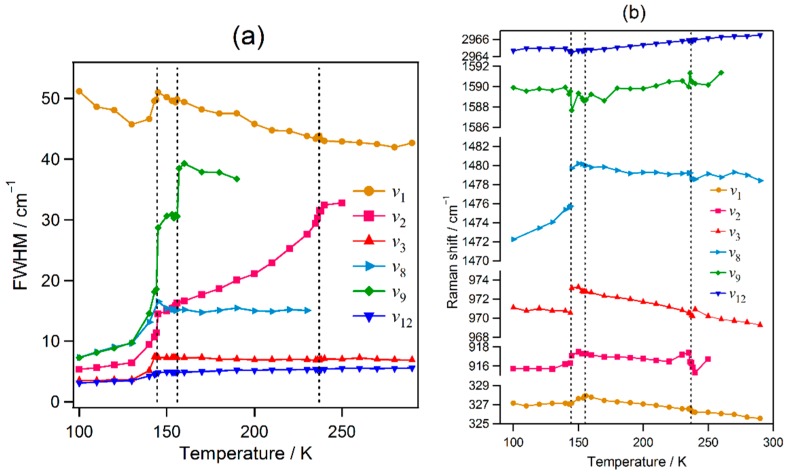
Temperature dependence of the (**a**) widths and (**b**) peak positions of the Raman bands of MAPbBr_3_.

**Figure 3 molecules-24-00626-f003:**
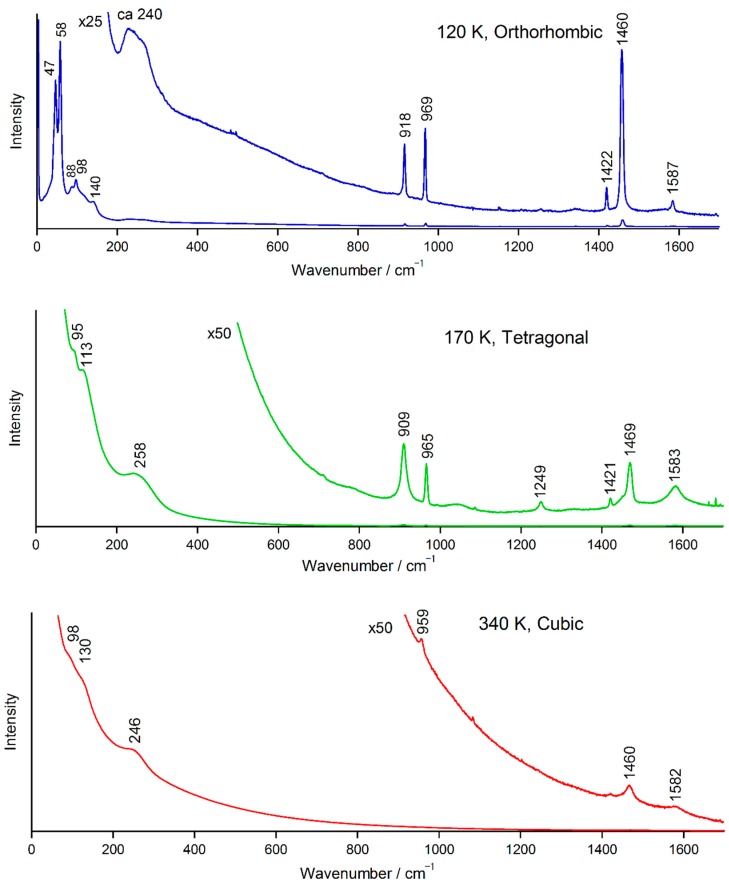
Raman spectra of a MAPbI_3_ pellet; excitation wavelength: 830 nm.

**Figure 4 molecules-24-00626-f004:**
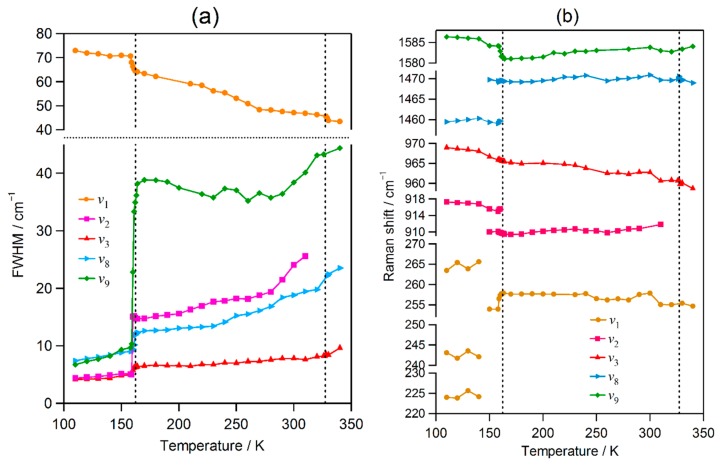
Temperature-dependent evolution of the (**a**) widths and (**b**) peak positions of the Raman bands of MAPbI_3_.

**Table 1 molecules-24-00626-t001:** Crystal structures [9,10,11] and transition temperatures [13] of CH_3_NH_3_PbX_3_ (X = I, Br).

**CH_3_NH_3_PbBr_3_**						
orthorhombic	————	tetragonal I	————	tetragonal II	————	cubic
*Pnma*: D_2h_ *Z* = 4	148.8 K	*P*4/*mmm*: D_4 h_ *Z* = 1	154.0 K	*I*4/*mcm*: D_4 h_ *Z* = 4	236.3 K	$Pm\bar{3}m$: O _h_ *Z* = 1
**CH_3_NH_3_PbI_3_**						
orthorhombic	—————————————————			tetragonal	————	cubic
*Pnma*: D_2h_ *Z* = 4		161.4 K		*I*4/*mcm*: D_4 h_ *Z* = 4	330.4 K	$Pm\bar{3}m$: O _h_ *Z* = 1

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
