# Peer review of "Temperature-Dependent Evolution of Raman Spectra of Methylammonium Lead Halide Perovskites, CH3NH3PbX3 (X = I, Br)"

_molecules, 2019, doi:10.3390/molecules24030626_

Round 1

Reviewer 1 Report

  The manuscript reports the temperature-dependent evolution of the Raman spectra of the hybrid perovskites MAPbI3 and MAPbBr3. The authors recorded the Raman spectra over a wide temperature range, calculated the full width at half maximum (FWHM) and monitored the evolution of the peak position of specific bands, in order to correlate them to the phase transitions. However there are already numerous in depth works, recently published on this specific topic (DOI: 10.1021/acs.jpcc.8b04669) and it is unclear what new information this work provides to the field. Furthermore the reported citations are pretty old, (2016 the most recent) which seems that the authors are missing important recent advances on this specific topic. Therefore, those results are interesting, but they are not novel, and thorough enough to be published in Molecules. They could be suitable for publication e.g. to Chemical Physics Letters.

Comments to the authors:

1.     I would recommend to the authors to study all the recent literature and revise their manuscript in a such a manner that they can present a novel work that can cover any missing information on this specific topic.

Author Response

Thanks very much for your comments, please find the response in the attachment.

Reviewer 2 Report

Presented manuscript deals with temperature dependence of MAPI and MAPBr crystals Raman spectra. This topic is in general very interesting for the community, but it means that the number of papers dealing with subject is high. Therefore authors should include more papers into reference list, my recommendations are here:

https://pubs.acs.org/doi/ipdf/10.1021/acs.jpcc.8b04669

J. Phys. Chem. Lett. (2016), 7 (3) 7 529-535

But there is more much more relevant studies. Mainly the first JPCC paper is very similar, measurement with 830 nm excitation, to even lower temperatures. The main difference is detail interpretation of the vibration modes based on numerical simulations. Assignment of the bands in this manuscript is not always consistent, for example the 1420 cmr. Is there a good reason for this difference? Moreover the spectra related to the organic cation does not differ a lot with changing halogen in the lattice, it seems that the part at low Raman shifts is more relevant and interesting (as the case of ν1 band).

To conclude, it is very nice measurement/set of experimental data. But is there any additional value for the reader, something beyond the description how the effect looks like? In my eyes authors should improve the discussion part before publishing.

Author Response

(The authors gave the same response as above.)

Reviewer 3 Report

The authors present a Raman study of two hybrid-organic perovskite materials prepared as pellets. They inspected the development of the lines that are mainly involve MA ionic vibrations with temprature. Interestingly, they see aprupt changes for the first-order phase transition and no such behaviour for the high-temperature transition which is expected to be of second order.  Another interesting finding is that the line of the MA torsional vibration gets narrower with increasing temperature for both materials.

The paper should be improved by a more substantial discussion of the measurements, the data analysis, and the results:

Why the motional narrowing is occuring?

What is seen at the temperatur of the second phase transition and what does it mean?

What is the reason of the fluctuations seen in Fig. 2 and 4. Are they larger than the errors? Error bars should be shown in Fig. 2 and 4. And it should be explained how the FWHH and the peak postions have been extracted from the spectra.

The Raman spectra show a more or less strong background. How it is considered in the data analysis and why it is present?

What about sample damage? What is the grain size in the pellets? What is the impurity content of the samples and is it constant for the various measurements?

Author Response

(The authors gave the same response as above.)

Round 2

Reviewer 1 Report

After the inclusion of all reviewers’ comments the manuscript looks consistent and up to date. I recommend to be published as it is.

Reviewer 2 Report

The paper quality is improved after the revisions. But I still do not see strong scientific message, something why to cite this paper. On the other hand, there is a well prepared summary of Raman(T) measurement, may be interesting for newcommers to the field. I do not have any other comments to the manuscript.

Reviewer 3 Report

The paper has been revised according to my comments. Doubling of text (line 145-147 and  194 to 196) is not necessary.